# The Impact of Information Infrastructure on Air Pollution: Empirical Evidence from China

**DOI:** 10.3390/ijerph192114351

**Published:** 2022-11-02

**Authors:** Pei Zhang, Peiran Chen, Fan Xiao, Yong Sun, Shuyan Ma, Ziwei Zhao

**Affiliations:** 1Institute of Geographic Sciences and Natural Resources Research, Chinese Academy of Sciences, Beijing 100101, China; 2School of Public Administration, Guangzhou University, Guangzhou 510006, China; 3School of Public Policy and Management, University of Chinese Academy of Sciences, Beijing 100049, China

**Keywords:** information infrastructure, air pollution, industrial structure upgrading, technology innovation, China

## Abstract

Information infrastructure construction has become an essential support for the new global technological revolution and industrial change. To examine whether information infrastructure can mitigate the level of air pollution, this paper measures the development level of information infrastructure in each region using the entropy-TOPSIS method based on the data of 31 Chinese provinces from 2013 to 2020. On this basis, it explores the impact of information infrastructure on atmospheric pollution and its mechanism using spatial measures and mediating effects. The results show that: (1) Information infrastructure can effectively improve air quality, though its spatial spillover effect is not obvious. (2) In addition to directly reducing air pollution, information infrastructure can also improve air quality by influencing industrial structure upgrading, or by influencing technological innovation first and then industrial structure upgrading. By exploring the impact of information infrastructure on air pollution and its action path, this paper expects to provide some scientific reference value for the construction of information infrastructure under the background of the new global technological revolution.

## 1. Introduction

In recent years, Chinese government departments have attached great importance to the prevention and control of air pollution. They have successively issued the Ambient Air Quality Standards, the Action Plan for the Prevention and Control of Air Pollution, the Three-Year Action Plan for Winning the Blue Sky Defense War and other related governance measures and strengthened law enforcement, aiming to continuously improve China’s air quality. According to the China Ecological Environment Status Bulletin 2021, the average concentration of PM2.5 in 339 cities nationwide was 30 µg/m^3^ in 2021, down about 40% from 2015; however, more than 1/3 of the cities still failed to meet air quality standards [1], which also means that the situation of air pollution management is still severe in China. However, in today’s era of Industry 4.0, with the development of machine learning, Internet of Things (IoT), Big Data, and augmented reality [2], information infrastructure is more capable of breaking the bottleneck of air pollution control and achieving air quality improvement than transportation infrastructure that achieves air pollution reduction under virtue of the spatial and temporal compression effect [3,4]. Therefore, it is of great theoretical and practical significance to explore the relationship between information infrastructure and air quality improvement.

Numerous scholars have conducted studies on air pollution from different perspectives. the results found that PM2.5 is one of the most harmful substances affecting air quality [5,6]. The factors affecting the increase of PM2.5 concentration can be attributed to natural and socio-economic factors, including the level of economic development, industrial structure, environmental regulations, urbanization level, or natural factors [6,7,8,9]. Although some scholars are aware of the possible impact of information infrastructure on air pollution, there is still a debate on its emission reduction effect; three views have been formed. First, the construction of information infrastructure will aggravate air pollution; for example, Sadorsky [10] and Park [11] argue that the manufacturing and use of ICT equipment consume large amounts of energy, exacerbate the country’s heavy dependence on fossil fuels, and induce carbon-intensive intermediate inputs in the electricity and basic materials sectors [12]. The disposal of used electronic equipment can also exacerbate environmental pollution levels. Second, information infrastructure can improve air quality, as Wu et al. [13] argued that the construction of information infrastructure can achieve intelligent environmental governance, and reduce the degree of resource mismatch and the level of environmental pollution. At the same time, information technology, such as the Internet, can enhance learning and communication among economic agents, accelerate the diffusion and application of advanced technologies, and promote the transformation and upgrading of industrial structures and green economic development [14]; so, countries with weak information infrastructure construction do not play their role in reducing environmental pollution [15]. Third, there is a non-linear relationship between information infrastructure construction and air pollution. On the one hand, information infrastructure promotes the development of information technology and digitalization [16]. Information and communications technology (ICT) can achieve energy saving and emission reduction through the development of smart cities, transportation systems, and industrial process optimization [17]. Digitization allows the opportunity to increase the energy efficiency of energy companies [18,19] and plays a central role in the transition of the energy sector [20], improving green total factor energy efficiency [21]. On the other hand, the application of ICT and digitization requires a large amount of energy and materials, as well as the continuous elimination of electronic waste, and will negatively affect the quality of the environment. Thus, the relationship between information infrastructure and environmental pollution shows an inverted U-shape. Although the existing studies on the impact of information infrastructure on air pollution are getting better, there is still a lack of exploring the relationship between information infrastructure and air quality and its mechanism of action from the perspective of spatial spillover in the context of a new generation information infrastructure construction.

Against this background, this paper attempts to explore whether information infrastructure can improve air quality. If it can, does it have a spatial spillover effect on air quality improvement based on the connectivity of information technology and the mobility of air pollution? The answers to these questions not only help to assess the environmental governance effects of information infrastructure construction but also have important reference significance in promoting synergistic pollution prevention and control among Chinese regions and promoting China’s high-quality economic development.

The innovations of this paper are as follows: first, the black box of the role of information infrastructure on air quality improvement is unveiled from the national scale by including information infrastructure construction and air quality improvement into a unified research framework; second, the spatial spillover effect of information infrastructure construction on air quality improvement is explored to provide a more precise development direction for the joint prevention and control of regional air pollution from the perspective of information infrastructure construction.

## 2. Theoretical Framework and Research Hypothesis

Based on established research and relevant underlying theories, this section emphasizes the important role of ICT-led information infrastructure in mitigating air pollution and proposes corresponding research hypotheses. It mainly includes two parts: firstly, the relationship between information infrastructure and air pollution; secondly, the mechanism of information infrastructure affecting air pollution. 

### 2.1. Information Infrastructure and Air Pollution

Numerous studies have concluded that information infrastructures increase air pollution. First, information infrastructure exacerbates greenhouse gas emissions. Ge et al. state that 5G, big data, cloud computing, and other information infrastructures emit 2% of the global total [22]; this finding is consistent with the Global e-Sustainability Initiative and Buttazoni [23]. Second, as the integration rate of information infrastructures, such as 5G base stations, and hardware facilities, such as computers, continues to increase, the large-scale use of electrical equipment will increase power and energy consumption [24], thus exacerbating environmental pollution. Data show that energy consumption in the information and communication sector accounts for more than 10% of the total global energy consumption [23]. Third, some scholars point out that the network is always a prerequisite for information infrastructure applications. If network security is not ensured, the risk of leakage of confidential information on the shared Internet will increase [25,26]. This makes some polluting companies reluctant to purchase intelligent pollution control equipment. Some chemical companies with high pollution emissions may not be able to warn the pollution defense in time by smart devices and the air quality may face great challenges [23].

Nevertheless, the role of information infrastructure in improving air quality cannot be ignored. On the one hand, the Chinese government proposed the strategy of “integrated development of informatization and industrialization” in 2002, aiming to promote the green transformation of traditional industries through ICT (information and communication technologies) and other technologies. Therefore, by promoting the development of new industries and upgrading industrial structures, the construction of information infrastructure will further promote the agglomeration of new industries based on the Internet, thus reducing the proportion of traditional high-pollution industries and curbing air pollution. Some scholars have confirmed that the combination of the two plays a positive role in energy management and carbon emissions [27,28,29]. On the other hand, information infrastructure can effectively promote technological advances in production technologies, energy conservation technologies, and environmental conservation technologies [29], thus contributing to the mitigation of air pollution. In addition, the construction of information infrastructure makes all kinds of sensors, intelligent monitoring equipment, and other intelligent factors applied to the production activities of enterprises; optimizes and upgrades the enterprise pollution management mode and technical means in all aspects; collects in real-time and dynamically various resource information such as air, water, and energy closely related to the enterprise emission activities; and intelligently senses and automatically controls the environmental pollution, energy consumption, and ecological damage, etc. The traditional method of sewage discharge and management of enterprises is thus changed into a smart management mode. Based on the above analysis, the following hypothesis is proposed.

**Hypothesis 1.** *Information infrastructure can mitigate air pollution levels*.

At the same time, the cross-regional property of information infrastructure may make it exhibit spatial spillover effects. On the one hand, there is an interaction between the development of information infrastructure between regions; the construction of local information infrastructure will improve the level of air quality in the neighboring regions by influencing their information infrastructure construction. On the other hand, connectivity, permeability, integration, and synergy are the main attributes of information infrastructure. Under the role of information infrastructure, regional spatial structure is constantly reshaped, and elements in different spaces can break through the limitation of geographical distance to quickly reorganize and efficiently link up to achieve cross-regional division of labor and cooperation. Economic linkage and cross-border pollution are the main reasons for the spatial correlation of air pollution. Industries and markets in various regions are highly interrelated. Economic activities in one region will affect the economic activities in the regions with which they are economically linked, thus having an impact on the air pollution level in the region. In the process of information infrastructure development, resources are guided beyond administrative divisions for optimal allocation. Spatial correlation of technological innovation capabilities is generated among economic agents from different regions, while information transparency reinforces the neighborhood and competition effects of technological innovation, resulting in a spatial spillover of technological innovation levels. Thus, the construction of local information infrastructure can indirectly lead to the optimization of resource allocation and technological innovation levels in neighboring areas, improving the air pollution level in neighboring areas. In this regard, this article proposes the following research hypothesis:

**Hypothesis 2.** *There is a spatial spillover effect of information infrastructure on air pollution mitigation*.

### 2.2. Mechanisms by Which Information Infrastructure Affects Air Pollution

The information infrastructure itself contains technological innovation. The Porter hypothesis suggests that technological innovation can improve environmental quality by enhancing the productivity of enterprises and offsetting the cost of environmental protection [30]. Specifically, big data, cloud computing, the Internet of Things, and other technical platforms derived from the operation of information infrastructure can effectively monitor consumer behavior and producer behavior and build a large, dynamic database. Economic agents make rational decisions through the analysis and processing of big data contents; in this process, consumers become the driving force of technological innovation and the consumption behavior and consumption bias of many consumers point out the direction for technological innovation and strengthen the motivation of economic agents for technological innovation. At the same time, the development of information technology can effectively reduce the communication cost of collaborative innovators, promote the diversification of technological innovation subjects, and improve collaborative innovation performance by sharing innovation resources with complementary advantages among collaborative innovators. With the improvement of technological innovation level, economic agents produce clean and non-polluting products by improving the efficiency of resource utilization, thus reducing the pollution emission in the up-production process. Therefore, the following hypothesis is proposed:

**Hypothesis 3.** 
*Information infrastructure mitigates air pollution through technological innovation.*


Meanwhile, the construction of information infrastructure will mitigate air pollution by promoting industrial structure upgrading. On the one hand, the construction of information infrastructure promotes the digitization of industry through technological innovation, which effectively improves the traditional production model and industrial organization model [31,32]. The relatively lower product prices after the digitization of industries will promote the transfer of production factors such as capital and labor to higher sectors, thus promoting the upgrading of industrial structures. On the other hand, information infrastructure construction promotes digital industrialization by driving the widespread application of new-generation information technology to form new business models and new goods or services [33]. The newly created products or services drive the transformation of the consumption structure [34], giving rise to diversified service forms, forming the multiplier effect of digital industrialization, and further promoting the upgrading of industrial structure. Along with the upgrading of industrial structure, industries with high pollution and high energy consumption will be eliminated, and new industries and modern service industries will accelerate their development and growth. New industries will use more non-polluting and clean production factors for production, and emissions of pollutants such as sulfur dioxide and soot from industrial production will be reduced, thus improving air quality. Based on the above analysis, the following hypotheses are proposed.

**Hypothesis 4.** *Information infrastructure mitigates air pollution by promoting industrial structure upgrading*.

**Hypothesis 5.** *Information infrastructure first influences technological innovation and then drives industrial structure upgrading, thereby slowing down air pollution*.

In summary, this study constructs a theoretical framework for the impact of information infrastructure on air pollution (Figure 1).

## 3. Study Design

### 3.1. Study Area and Data Acquisition

This paper takes mainland China (excluding Hong Kong, Macao, and Taiwan) as the study area; 31 provincial administrative units were selected as the study objects. Among them, the data of information infrastructure-related indicators (e.g., the number of industrial Internet-related enterprises, etc.) are mainly obtained from the enterprise search platform (https://www.qcc.com/, accessed on 6 January 2022). PM2.5 data is calculated based on the annual average global PM2.5 concentration data provided by the Center for International Earth Science Information Network (CIESIN) at Columbia University (http://sedac.ciesin.columbia.edu, accessed on 1 August 2022). Other data such as the level of economic development, population density, and green area were obtained from the China Statistical Yearbook (2013–2020).

### 3.2. Methods

#### 3.2.1. Entropy-TOPSIS Method

This paper uses the Entropy-TOPSIS method to measure the level of information infrastructure development. The Entropy-TOPSIS method is an improvement of the traditional TOPSIS evaluation method [35,36,37]. It first determines the weights of evaluation indexes through the entropy value method, which avoids the influence of subjective factors. Then, it determines the ranking of evaluation objects through the TOPSIS method using the technique of approximating ideal solutions. The calculation steps are as follows.

① Assuming that there are m evaluated objects and *n* evaluation indicators for each evaluated object, a judgment matrix is constructed as follows:(1)$$X={\left(x_{ij}\right)}_{m\times n}\;\left(i=1,\;2,\cdots m;j=1,\;2,\cdots n\right)$$

② The judgment matrix is normalized and the indicators in this paper are all positive, thus:(2)$$Y_{ij}=\frac{X_{ij}-\min\;\left(X_{i}\right)}{\max\left(X_{i}\right)-\min\left(X_{i}\right)}$$

③ Calculating information entropy:(3)$$E_{j}=-k\sum_{i=1}^{m}p_{ij}lnp_{ij}$$

In the formula, $p_{ij}=\frac{Y_{ij}}{\sum_{i=1}^{m}Y_{ij}}$; $k=\frac{1}{lnm}$.

④ Determine the weights of each indicator. The weights of each indicator are calculated by information entropy.
(4)$$w_{j}=\frac{1-E_{j}}{\sum_{j=1}^{n}\;\left(1-E_{j}\right)}$$

In the formula, $w_{j}\in \;\left[0,1\right]$, and $\sum_{j=1}^{n}\;\left(w_{j}\right)=1$.

⑤ Calculating the weighting matrix:(5)$$R={\left(r_{ij}\right)}_{m\times n},\;r_{ij}=w_{j}\times x_{ij}\;\left(i=1,2,\cdots ,m;j=1,2,\cdots ,n\right)$$

⑥ Determine the optimal solutions ${s_{j}}^{+}$ and ${s_{j}}^{-}$:(6)$${s_{j}}^{+}\max\;\left(r_{1j},r_{1j},\cdots ,r_{nj}\right),\;{s_{j}}^{-}\max\;\left(r_{1j},r_{1j},\cdots ,r_{nj}\right)$$

⑦ Calculate the Euclidean distances of various solutions from the optimal and inferior solutions.
(7)$${sep_{i}}^{+}\sqrt{\sum_{j=1}^{n}\;{\left(s_{j}{}^{+}-r_{ij}\right)}^{2}},\;sep_{i}{}^{-}\sqrt{\sum_{j=1}^{n}\;{\left(s_{j}{}^{-}-r_{ij}\right)}^{2}}$$

⑧ Calculation of the composite evaluation index.
(8)$$C_{i}=\frac{sep_{i}{}^{-}}{sep_{i}{}^{+}+sep_{i}{}^{-}},\;C_{i}\in \;\left[0,1\right]$$

#### 3.2.2. Global Spatial Autocorrelation

The global spatial autocorrelation is used to describe the clustering characteristics of the observed variables in the whole space, which is usually measured by the global Moran’s I. It can reveal the similarity of air quality in neighboring areas; its calculation formula is as follows [38]:(9)$$I=\frac{\sum_{i=1}^{n}\sum_{j=1}^{n}W_{ij}\;\left(x_{i}-\bar{x}\right)\;\left(x_{j}-\bar{x}\right)}{\sum_{i=1}^{n}\sum_{j=1}^{n}w_{ij}}$$

In the formula, I denotes the global Moran index, taking values between −1 and 1. I greater than 0 indicates a positive spatial correlation, that is, high-high and low-low clusters. I less than 0 indicates a negative spatial correlation, that is, high-low and low-high outliers. I close to 0 indicates the random distribution and no spatial autocorrelation. n is the number of provinces. $x_{i}\left(x_{j}\right)$ is the $i\;\left(j\right)$ place of air pollution level. $W_{ij}$ denotes the adjacency (0–1) spatial weight matrix. $S^{2}$ is the sample variance, $w_{ij}$ is determined by the adjacency (0–1) spatial weight matrix, when place i is adjacent to place j, $w_{ij}$ takes the value of 1 and 0 if it is not adjacent.

#### 3.2.3. Space Durbin Model

When analyzing the impact of information infrastructure on air pollution, the OLS (ordinary least squares) method will often lead to biased regression results if the spatial correlation effects among provinces are ignored. However, compared with the Spatial Lag Model (SLM) and the Spatial Error Model (SEM), the Spatial Durbin Model (SDM) not only considers the endogenous and exogenous spatial interaction effects but also reflects the spatial hysteresis of explanatory variables and explained variables. Therefore, the Spatial Durbin Model is used in this paper, which is calculated as follows.

Equation (10) is the spatial Durbin econometric model derived by Anselin [39], based on Durbin’s residual autocorrelation time series model inference.
(10)$$\left(I_{n}-\rho W\right)y=\left(I_{n}-\rho W\right)X\beta +\varepsilon$$
(11)y=ρWy+Xβ−ρWXβ+ε

The above equation can also be expressed in Equation (12).
(12)$$y=X\beta +{\left(I_{n}-\rho W\right)}^{-1}\varepsilon$$

The spatial Durbin measure model can be represented by Equation (13), where *y* is a vector of n×1 columns of observations of the dependent variable and X is a n×k order matrix of k explanatory variable observations. $\beta_{1}$ is a vector of k×1 order regression coefficients. W is the spatially lagged dependent variable. ρ is the spatially lagged autoregressive parameter, and the spatial matrix WX is a spatially lagged explanatory variable added to the model. $\beta_{2}$ is its regression coefficient vector, which indicates the effect of neighboring regional variables on the dependent variable.
(13)$$y=\rho Wy+X\beta_{1}-\rho WX\beta_{2}+\varepsilon$$

#### 3.2.4. Intermediary Effect Model

Referring to the procedure of testing the mediating effect by Wen [40], it is found that the test of mediating effect is mainly using stepwise regression analysis, which consists of three steps. First, the overall effect of the core explanatory variables on the explanatory variables is examined, and it is necessary to ensure that its effect is significant. Second, the effect of the core explanatory variables on the mediating variables is examined. The third is the need to examine the effect of the core explanatory variables and the mediating variables on the explanatory variables simultaneously. If the influence coefficient of core explanatory variables is lower than step 1 after the inclusion of mediating variables, and the influence of mediating variables is significant, the mediating variables play a mediating role. Additionally, the size of the mediating effect can be calculated based on the correlation coefficient; the product of the coefficient of influence of the core explanatory variables on the mediating variables in step 2 and the coefficient of influence of the mediating variables on the explained variables in step 3 is the size of the mediating effect. Therefore, this paper uses stepwise regression analysis to sequentially advance the exploration of the path of information infrastructure affecting air pollution according to the testing procedure of the mediating effect model.

Step 1, a base model is constructed to examine the overall effect of information infrastructure construction on air pollution.
(14)$$POLL_{it}=\alpha +\beta *INFOR_{it}+\beta_{1}*CONT_{it}+\mu_{it}$$
where i denotes cross section, t denotes time, POLL denotes air pollution, INFOR denotes the level of information infrastructure development, CONT denotes the remaining relevant control variables, α is a constant term, and $\mu_{it}$ denotes a random disturbance term.

In step 2, examine the effect of the level of information infrastructure development on the mediating variable (MED).
(15)$$MED_{it}=\beta_{2}+\beta_{3}*INFOR_{it}+\beta_{4}*CONT_{it}+\varepsilon_{it}$$

In step 3, the information infrastructure and mediating variables are jointly included in the air pollution regression model to test for mediating effects.
(16)$$POLL_{it}=\beta_{5}+\beta_{6}*INFOR_{it}+\beta_{7}*MED_{it}+\beta_{8}*CONT_{it}+\mu_{it}$$

### 3.3. Variable Selection

#### 3.3.1. Explained Variable

The explanatory variable in this paper is the air pollution level (*POLL*). The air pollution problem in China is mainly PM2.5 pollution. It is estimated that the annual number of deaths attributable to PM2.5 pollution in China has increased to 971,000 as of 2017 [41]. Compared with other atmospheric particulate matter, PM2.5 presents greater health risks and hazards due to its small particle size, large surface area, and its tendency to carry airborne toxins [42]. Therefore, it has been used by a large number of scholars to measure air pollution levels [43,44,45]. Based on this, PM2.5 is selected as a proxy variable for air pollution levels in this paper, and its larger value indicates poorer air quality.

#### 3.3.2. Explanatory Variable

The core explanatory variable is the level of information infrastructure development (*INFOR*). According to the definition of information infrastructure by the National Development and Reform Commission of China on 20 April 2020, it includes communication network infrastructure represented by 5G, the Internet of Things, Industrial Internet, and satellite Internet; new technology infrastructure represented by artificial intelligence, cloud computing, and blockchain; and computing power infrastructure represented by data centers and intelligent computing centers. Referring to the study of Qiao [23], however, patent data cannot scientifically evaluate the development scale of each type of information infrastructure and its true supply and demand level. The higher the development level of a certain type of information infrastructure in a region, the stronger supply and demand capacity of the region for the related facilities, and the number of related enterprises is naturally high. Therefore, this paper adopts the number of relevant enterprises of corresponding facilities in the region as a proxy variable (Table 1).

The control variables include (Table 2.): ① Industrial structure upgrading (*ISU*). Industrial structure upgrading is manifested by the continuous optimization of factor input structure, the flow of resources to higher productivity sectors, and the gradual replacement of low-value-added and high-polluting industries by high-value-added and green industries. This further leads to a decrease in total energy consumption and thus contributes to the mitigation of air pollution levels. In this paper, we refer to Song et al. [46] to construct an industrial structure upgrading index (*ISU*) for Chinese provinces and cities to investigate the mediating effect of information infrastructure on air pollution.
(17)$$ISU_{it}=\overset{3}{\underset{n=1}{\sum }}Y_{int}*n,\;n=1,\;2,\;3$$

Here, $Y_{int}$ indicates the proportion of total output value of nth industry in region i to regional GDP in year t. 1≤ISU≤3, the value reflects the evolution of industrial structure from primary industry to secondary and tertiary industries. The closer its value is to 3, the higher the level of industrial structure upgrading.

② Technological innovation (*TECH*). According to the Porter hypothesis [30], technological innovation can improve environmental quality. Technological innovation is reflected in the innovation of pollution control technology and production technology; pollution control technology improves the efficiency of pollution control and plays the role of “bottoming out” of terminal control. Innovation in production technology improves the efficiency of resource utilization and reduces the total amount of pollution generated, thus improving air pollution. Since patent data is a rare measure of technological innovation [47], to truly reflect the level of technological innovation in Chinese provinces, this paper uses the ratio of the number of licenses to the number of applications to measure technological innovation.

③ The level of economic development (*PGDP*). Economic growth itself does not improve the environment; the reason for the reduction of air pollution is that after a certain level of economic development, the government will increase environmental management, thus improving environmental quality [48]. Referring to the study of Qiao [23], GDP per capita is chosen to measure the level of economic development in this paper.

④ The squared term of the level of economic development (*PGDP × PGDP*). To test the EKC hypothesis [49], the squared term of the level of economic development is added to the model in this paper.

⑤ Population density (*DPOP*). Population density is closely related to air pollution; the higher the population density, the more serious the air pollution in the area [50,51]. This is because the increase in air pollutant emissions is largely due to the increasingly frequent social activities of humans; the increase in population density is an important reason for the more frequent social activities of people. The increase in population density will undoubtedly increase basic needs such as housing and travel, which will bring more road dust and vehicle emissions. Moreover, the increase in population density will make commercial behaviors such as leisure and entertainment more and more frequent, indirectly promoting the construction of large shopping malls and the resulting traffic pressure, while more and more commercial individuals will in turn stimulate people’s commercial behaviors. It follows that emissions of air pollutants caused by human activities will continue to increase with the rise in population density. Therefore, drawing on the studies of Liu [50] and Jiang [51], the ratio of regional population to the regional area is chosen to measure population density in this paper.

⑥ Urbanization process (*CITY*). Urbanization brings about the “living effect”. On the one hand, the air pollution problems caused by human life have become more intense, such as the rapid growth of the number of motor vehicles resulting in serious pollution from vehicle exhaust. On the other hand, urbanization has further promoted industrialization; air pollution from production activities has become more concentrated in urban areas. Of course, urbanization also brings the “production effect”, and the increase in urbanization is also conducive to the environmental protection department to give full play to the role of pollution treatment facilities to centralize the treatment of air pollution. Therefore, urbanization affects air quality. Among the various indicators, the indicator of “urban population to total population” is widely accepted and has authoritative data sources [52]. Therefore, in this paper, the proportion of the urban population is chosen to measure the urbanization process.

⑦ Greening level (*GREE*). The level of greenery may have both positive and negative effects on air quality [53]. On the one hand, green areas can act as carbon sinks to clean the air because the dust retention effect of plant leaves has mechanisms to remove atmospheric particulate matter and promote the migration of atmospheric particulate matter to the soil environment. On the other hand, excessive investment in green space may also crowd out other aspects of environmental protection expenditure. Drawing on the study of Qiao [23], this paper uses green space per capita to measure the level of greening.

⑧ Government regulation (*GOV*). Kolstad argued that government intervention or regulation would contribute to environmental improvement [54]. Because government regulation can effectively curb environmental pollution by changing the production methods or production location of enterprises [55], air quality improvement is a direct reflection of the effect of government management. However, no matter what measures the government takes for management, it cannot be separated from the support of public finance. Therefore, this paper uses the ratio of energy conservation and environmental protection expenditure to public finance expenditure to measure the degree of government management.

## 4. Empirical Results

### 4.1. Impact of Information Infrastructure on Air Pollution

The results of spatial autocorrelation analysis proved that air pollution has a certain positive spatial correlation (Figure 2). Therefore, spatial dependence should be considered to explore its influencing factors, and spatial econometric models are appropriate. Before conducting the spatial econometric analysis, this paper conducted model screening of the LM test, the Hausman test, the LR test, and the Wald test (Table 3). The Spatial Durbin Model (SDM) under fixed effects was finally determined to explore the influence of information infrastructure on air pollution.

Based on the spatial adjacency weight matrix, this paper uses SDM under fixed effects to estimate the impact of information infrastructure on air quality; the results are shown in Table 4. Among them, model 1 (M1) adopts a time-fixed effect, model 2 (M2) adopts an individual-fixed effect, and model 3 (M3) adopts a double-fixed effect. The results show that the fit of model 1 is higher than that of model 2 and model 3, with stronger credibility (R^2^ = 0.441), so this paper focuses on the regression results of model 1. Meanwhile, to better reflect the direction and magnitude of spatial spillover of each influencing factor, this paper draws on LeSage [56] to derive the partial differential and thus estimate the direct and indirect effects of each influencing factor on air pollution based on the estimation results of the time-fixed effects SDM model with a better fit; the results are shown in Table 5.

Looking at the specific results (Table 4), the effect of the core explanatory variable Information Infrastructure (*INFOR*) on air pollution passed the 1% significance level test in the negative direction. This indicates that the development of information infrastructure can improve air quality. Hypothesis 1 was verified. Meanwhile, the spatially lagged term of information infrastructure (*W × INFOR*) has a significant positive effect on air pollution, indicating that information infrastructure in the surrounding areas has a positive transmission effect on local air pollution. In other words, the development of information infrastructure in the surrounding areas aggravates local air pollution. However, Table 5 shows that the indirect effect of information infrastructure on air pollution does not pass the significance level test, indicating that local air pollution is not significantly influenced by the information infrastructure in the surrounding areas. Thus, it seems that the spatial spillover effect of information infrastructure on air quality improvement is not significant. Thus, Hypothesis 2 is not valid.

The industrial structure upgrading (*ISU*) has a significant negative effect on air pollution. The impact of industrial structure upgrading (*ISU*) and its spatial lag (*W × ISU*) on air pollution pass the 1% significance level test (Table 4). Meanwhile, combined with the estimation results in Table 5, both the direct and indirect effects of industrial structure upgrading on air pollution pass the 1% significance level test, where the direction of the direct effect is negative and the direction of the indirect effect is positive, which is consistent with the results in Table 4. It shows that the industrial structure upgrading can slow down the air pollution level. However, the improvement effect mainly comes from the local area, and the industrial structure upgrading in the surrounding areas will aggravate the local air pollution instead. The reason may be that the upgrading of the local industrial structure will transfer the polluting lower-level industries to the surrounding areas at the same time.

The effect of technological innovation (*TECH*) on air pollution passed the 1% significance level test with a negative direction while its spatial lag term (*W × TECH*) on air pollution did not pass the significance level test (Table 4), which is consistent with the results in Table 5. It indicates that technological innovation can reduce the air pollution level; the improvement effect mainly comes from the local area. The impact of technological innovation on air pollution in the surrounding areas is not significant. The reason may lie in the existence of technical barriers between regions, which prevent the technology spillover effect from being realized.

In addition, the variables of economic development level (*PGDP),* the square of economic development level (*PGDP*), population density (*DPOP*), urbanization process (*CITY*), and green area (*GREE*) passed the 1% significance level test while the degree of government governance (*GOV*) did not pass the significance level test (Table 4 and Table 5). This also verifies the scientific and rational nature of the selection of control variables in this study.

### 4.2. Pathways through which Information Infrastructure Affects Air Pollution

The above paper mainly examined the impact of information infrastructure on air pollution. As explained in the previous section, information infrastructure may affect technological innovation, industrial structure upgrading, and then air pollution. At the same time, information infrastructure may affect technological innovation first and then industrial structure upgrading, which in turn affects air pollution. To verify these possibilities, this paper further analyzes the impact paths of information infrastructure acting on air pollution through the mediating effects model proposed by Baron et al. [57].

Table 6 reports the test results of the mediating effects model. Column (1) shows the results of the baseline regression without the two mediating variables of technological innovation and industrial structure upgrading. Column (2) shows the regression results for the mediating variable technological innovation and the core explanatory variable information infrastructure, which has a significantly positive coefficient, indicating that information infrastructure promotes technological innovation. Column (3) shows the regression results of the explanatory variable air pollution, the mediating variable technological innovation, and the core explanatory variable information infrastructure. The coefficient of information infrastructure is significantly negative, while the coefficient of technological innovation does not pass the significance level test. It indicates that information infrastructure reduces air pollution but the coefficient of technological innovation is not significant. Meanwhile, the combined regression results of column (1), column (2), and column (3) indicate that information infrastructure affects technological innovation and air pollution, but there is no mediating effect of technological innovation in the process of information infrastructure acting air pollution, so Hypothesis 3 is not valid.

Column (4) shows the regression results of the mediating variable industrial structure upgrading and the core explanatory variable information infrastructure. Column (5) shows the regression results of the explanatory variable air pollution, the mediating variable industrial structure upgrading, and the core explanatory variable information infrastructure. Among them, column (1) shows that the effect of information infrastructure on air pollution is significantly negative. Column (4) shows that the effect of information infrastructure on industrial structure upgrading is significantly negative. Column (5) shows that when information infrastructure and industrial structure upgrading are put into regression, the impact of industrial structure upgrading on air pollution is significantly negative, while the impact of information infrastructure on air pollution is still significantly negative. The combined columns (1), (4), and (5) show that information infrastructure affects industrial structure upgrading first, then affects air pollution; there is a partial mediating effect of industrial structure upgrading. Therefore, both Hypothesis 1 and Hypothesis 4 are assumed to be valid.

Column (6) regresses the results of the mediating variable industrial structure upgrading, the mediating variable technological innovation, and the core explanatory variable information infrastructure. The coefficient of technological innovation passes the significance level test, while the coefficient of information infrastructure does not pass the significance level test. The combined columns (4), (2), and (6) show that technological innovation is the mediating variable of information infrastructure affecting industrial structure upgrading and there is a full mediation effect. Column (7) is the regression result of the explanatory variable air pollution and the mediating variable technological innovation, which indicates that the effect of technological innovation on air pollution is not significant. Column (8) is the regression result of the mediating variable industrial structure upgrading and the mediating variable technological innovation. The coefficient of technological innovation passes the significance level test, which again indicates that technological innovation will have a significant impact on industrial structure upgrading. Thus, it can be seen that information infrastructure affects technological innovation first, then affects industrial structure upgrading, and then affects air pollution; thus, Hypothesis 5 holds.

Column (9) shows the regression results of the explanatory variable air pollution, the mediating variable industrial structure upgrading, and the mediating variable technological innovation, where the coefficients of industrial structure upgrading and technological innovation pass the significance level test. Column (10) shows the regression results for the explanatory variable air pollution, the mediating variable industrial structure upgrading, the mediating variable technological innovation, and the core explanatory variable information infrastructure. Combining columns (1), (9), and (10), it again shows that industrial structure upgrading exercises a partial mediating effect in the process of information infrastructure affecting air pollution, which again verifies the validity of Hypothesis 4.

## 5. Conclusions and Discussion

### 5.1. Conclusions

This paper explores the impact of information infrastructure on air pollution and its action path, which has a very important theoretical and practical significance. It not only could provide theoretical support for the information infrastructure to improve air quality, but also might offer scientific reference for the construction and layout of information infrastructure in the context of the new round of global technological revolution. Based on the data from 31 provinces in China from 2013 to 2020, this paper measures the development level of information infrastructure in each province and region using the Entropy-TOPSIS weight method. On this basis, the impact of information infrastructure on air pollution and its mechanism of action are explored using spatial measurement and mediating effect methods. The main conclusions are as follows: (1) Information infrastructure can mitigate air pollution levels (Hypothesis 1 holds). (2) The spatial spillover effect of information infrastructure on air pollution is not obvious, i.e., the construction of local information infrastructure does not have a significant impact on air pollution in the surrounding areas (Hypothesis 2 does not hold). (3) Information infrastructure reduces air pollution levels by influencing industrial structure upgrading; industrial structure upgrading plays a mediating effect in the process of information infrastructure influencing air pollution (Hypothesis 3 holds). (4) The mediating effect of technological innovation in the process of information infrastructure affecting air pollution is not obvious (Hypothesis 4 does not hold), while information infrastructure mitigates air pollution levels by influencing technological innovation first and then industrial structure upgrading (Hypothesis 5 holds).

### 5.2. Discussion

The findings of this paper further confirm that the construction of information infrastructure can improve air quality and reveal its intrinsic mechanisms. Additionally, this paper tests that factors such as economic growth [49], population density [51], green area [23], and urbanization [58] can also affect air pollution. Combining the above findings, this paper draws the following policy insights: at the macro level, the national government can introduce relevant policies to encourage the construction of information infrastructure, such as financial subsidies, etc. At the micro level, each region should take into account its comparative advantages and promptly learn from the beneficial experiences of other provinces across the country, such as how information infrastructure can promote industrial structure upgrading, to actively improve air pollution levels.

Although this study accomplishes the initial objectives and clarifies the relationship between information infrastructure and air pollution, this paper still has some shortcomings. Due to the lack of information infrastructure data, the empirical study only explores the influence mechanism of information infrastructure on air pollution from the provincial level, while the influence mechanism is not the same in different spatial scales. It is an innovative idea to consider air pollution from the perspective of information infrastructure. However, air pollution is a complex issue. It may be more necessary to comprehensively analyze air pollution by integrating green technology (e.g., energy-efficient houses, emission control devices, etc. [59]), innovative materials (e.g., bamboo as green energetic plant [60,61], hydrogen [62], etc.). With the improvement of spatial and temporal data, it may become a future research direction to explore the mechanism of information infrastructure influence on spatial pollution based on multiple spatial scales.

## Figures and Tables

**Figure 1 ijerph-19-14351-f001:**
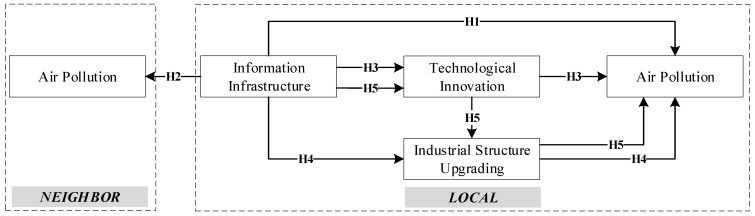
Conceptual framework of the impact of information infrastructure on air pollution.

**Figure 2 ijerph-19-14351-f002:**
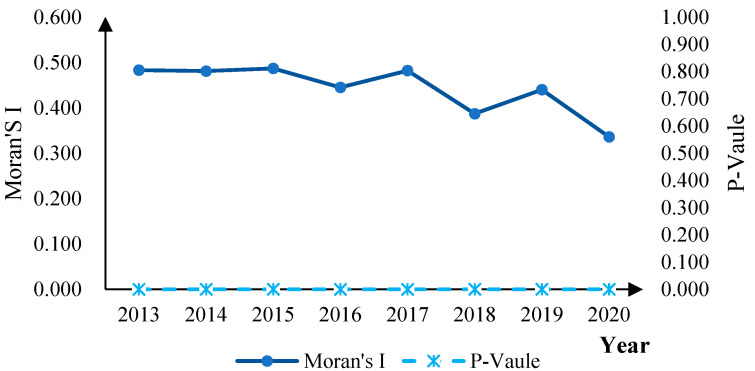
Global Moran’s I of air pollution levels.

**Table 1 ijerph-19-14351-t001:** Evaluation index system of development level of information infrastructure.

	First-Level Indicators	Second-Level Indicators
Information infrastructure	Communication network infrastructure (N1)	5G (N11) (N11) Internet of Things (N12) Industrial Internet (N13) Satellite Internet (N14)
	New technology infrastructure (N2)	Artificial Intelligence (N21) Cloud Computing (N22) Blockchain (N23)
	Algorithmic Infrastructure (N3)	Data Center (N31) (N31) Intelligent Computing Center (N32)

**Table 2 ijerph-19-14351-t002:** The descriptive statistical analysis results of main variables.

Variables	Mean	Std. Dev.	Min	Max	No. of Observations
*POLL*	36.351	14.663	4.414	85.629	248
*INFOR*	0.067	0.088	0.000	0.608	248
*ISU*	2.394	0.122	2.194	2.836	248
*TECH*	0.552	0.102	0.251	0.835	248
*PGDP*	0.591	0.277	0.220	1.649	248
*PGDP × PGDP*	0.425	0.458	0.049	2.719	248
*DPOP*	0.287	0.112	0.106	0.554	248
*CITY*	0.594	0.125	0.239	0.896	248
*GREE*	20.717	10.957	7.521	66.162	248
*GOV*	3.753	11.799	1.178	18.819	248

**Table 3 ijerph-19-14351-t003:** Test results of the spatial panel model.

Test Method	Test Results
LM_spatial_error	11.366 ***
LM_spatial_lag	31.797 ***
Hausman	100.12 ***
LR_sar	192.5 ***
LR_sem	168.63 ***
Wald_spatial_error	40.52 ***
Wald_spatial_lag	45.32 ***

Note. *** *p* < 0.01.

**Table 4 ijerph-19-14351-t004:** Regression results of the SDM.

Variables	Time-Fixed Effects (M1)	Individual-Fixed Effects (M2)	Mixed-Fixed Effects (M3)
*INFO*	−23.419 *** (0.00)	−8.523 (0.19)	2.203 (0.75)
*ISU*	−22.481 *** (0.00)	−30.070 *** (0.00)	−20.088 ** (0.02)
*TECH*	−26.237 *** (0.00)	−4.248 (0.12)	−8.450 *** (0.00)
*PGDP*	97.379 *** (0.00)	19.450 ** (0.01)	42.322 *** (0.00)
*PGDP × PGDP*	−41.786 *** (0.00)	−9.993 *** (0.00)	−18.426 *** (0.00)
*DPOP*	21.651 *** (0.00)	−8.763 (0.14)	−10.329 * (0.07)
*CITY*	107.311 *** (0.00)	72.269 *** (0.00)	127.522 *** (0.00)
*GREE*	−0.435 *** (0.00)	−0.238 ** (0.03)	−0.160 (0.14)
*GOV*	0.032 (0.45)	0.030 * (0.09)	0.028 * (0.10)
*W × INFO*	22.329 * (0.09)	1.508 (0.88)	13.802 (0.25)
*W × ISU*	36.241 *** (0.00)	−18.762 (0.31)	−15.400 (0.44)
*W × TECH*	7.914 (0.56)	3.767 (0.34)	−16.071 ** (0.01)
*W × PGDP*	−4.471 (0.83)	27.931 (0.10)	60.480 *** (0.00)
*W × PGDP × PGDP*	1.108 (0.93)	−18.489 ** (0.01)	−29.236 *** (0.00)
*W × DPOP*	−52.024 *** (0.00)	−7.122 (0.60)	−17.646 (0.20)
*W × CITY*	−49.096 ** (0.02)	−146.649 *** (0.00)	−106.837 *** (0.00)
*W × GREE*	−0.949 *** (0.00)	−0.583 ** (0.02)	−0.527 ** (0.04)
*W × GOV*	−0.187 * (0.09)	0.015 (0.75)	0.048 (0.29)
*Rho*	0.176 * (0.06)	0.043 (0.64)	−0.037 (0.72)
sigma2_e	52.995 *** (0.00)	8.837 *** (0.00)	7.727 *** (0.00)
Observations	248	248	248
R-squared	0.441	0.001	0.034
Number of Area	31	31	31

Note: *** *p* < 0.01, ** *p* < 0.05, * *p* < 0.1, standard errors are in parentheses.

**Table 5 ijerph-19-14351-t005:** The direct and indirect effect of independent variables on the air pollution.

Variables	Direct Effect	Indirect Effects	Total Effect
*INFO*	−22.350 *** (0.01)	21.265 (0.16)	−1.085 (0.95)
*ISU*	−21.526 *** (0.00)	36.198 *** (0.00)	14.672 (0.19)
*TECH*	−25.359 *** (0.00)	5.797 (0.73)	−19.563 (0.33)
*PGDP*	97.728 *** (0.00)	13.208 (0.58)	110.935 *** (0.00)
*PGDP × PGDP*	−41.934 *** (0.00)	−6.242 (0.65)	−48.176 *** (0.00)
*DPOP*	20.007 *** (0.00)	−54.567 *** (0.00)	−34.560 (0.12)
*CITY*	106.370 *** (0.00)	−33.644 (0.11)	72.726 *** (0.00)
*GREE*	−0.480 *** (0.00)	−1.167 *** (0.00)	−1.647 *** (0.00)
*GOV*	0.028 (0.50)	−0.211 (0.11)	−0.183 (0.21)

Note: *** *p* < 0.01, standard errors are in parentheses.

**Table 6 ijerph-19-14351-t006:** Results of stepwise regression mediating effect test.

Variables	(1)	(2)	(3)	(4)	(5)	(6)	(7)	(8)	(9)	(10)
	*POLL*	*TECH*	*POLL*	*ISU*	*POLL*	*ISU*	*POLL*	*ISU*	*POLL*	*POLL*
*INFO*	−28.497 * (0.07)	0.433 *** (0.00)	−28.093 * (0.08)	−0.085 * (0.06)	−32.579 ** (0.04)	−0.066 (0.19)				−31.341 ** (0.04)
*ISU*					−47.754 ** (0.01)				−44.814 ** (0.01)	−49.152 ** (0.01)
*TECH*			−0.933 (0.74)			−0.045 * (0.09)	−3.461 (0.27)	−0.051 ** (0.03)	−5.733 * (0.06)	−3.132 (0.23)
*PGDP*	−11.757 (0.55)	−0.095 (0.60)	−11.846 (0.55)	0.219 ** (0.02)	−1.309 (0.94)	0.215 ** (0.02)	−16.931 (0.42)	0.203 ** (0.03)	−7.852 (0.69)	−1.300 (0.94)
*PGDP × PGDP*	1.347 (0.89)	0.011 (0.88)	1.358 (0.89)	−0.057 * (0.08)	−1.396 (0.88)	−0.057 * (0.08)	0.897 (0.94)	−0.058 * (0.08)	−1.703 (0.88)	−1.441 (0.87)
*DPOP*	−6.801 (0.59)	0.043 (0.79)	−6.761 (0.60)	0.078 (0.42)	−3.057 (0.79)	0.080 (0.39)	−5.140 (0.71)	0.084 (0.37)	−1.369 (0.92)	−2.813 (0.81)
*CITY*	−77.347 ** (0.01)	0.496 * (0.05)	−76.884 ** (0.01)	0.876 *** (0.00)	−35.514 (0.30)	0.898 *** (0.00)	−79.727 ** (0.01)	0.892 *** (0.00)	−39.773 (0.22)	−32.735 (0.34)
*GREE*	−0.728 ** (0.02)	0.005 *** (0.00)	−0.724 ** (0.03)	0.003 *** (0.00)	−0.580 * (0.06)	0.003 *** (0.00)	−0.757 ** (0.02)	0.003 *** (0.00)	−0.611 * (0.06)	−0.560 * (0.08)
*GOV*	0.033 *** (0.00)	−0.000 *** (0.00)	0.033 *** (0.00)	−0.000 *** (0.00)	0.026 *** (0.00)	−0.000 *** (0.00)	0.038 *** (0.00)	−0.000 *** (0.00)	0.032 *** (0.00)	0.026 *** (0.00)
*Constant*	107.467 *** (0.00)	0.164 (0.25)	107.620 *** (0.00)	1.689 *** (0.00)	188.112 *** (0.00)	1.696 *** (0.00)	112.231 *** (0.00)	1.707 *** (0.00)	188.727 *** (0.00)	190.988 *** (0.00)
*Observations*	248	248	248	248	248	248	248	248	248	248
*R-squared*	0.709	0.163	0.709	0.748	0.740	0.753	0.686	0.750	0.714	0.742

Note: *** *p* < 0.01, ** *p* < 0.05, * *p* < 0.1, standard errors are in parentheses.

## Data Availability

The data presented in this study are available on request from the corresponding author. The data are not publicly available in accordance with consent provided by participants on the use of confidential data.

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
