# Peer review of "The Impact of Information Infrastructure on Air Pollution: Empirical Evidence from China"

_ijerph, 2022, doi:10.3390/ijerph192114351_

Round 1
Reviewer 1 Report
Information infrastructure construction has become essential support for the new global technological revolution and industrial change. this paper measures the development level of information infrastructure in each region using the entropy-TOPSIS method based on the data from 31 Chinese provinces from 2013 to 2020. On this basis, it explores the impact of information infrastructure on atmospheric pollution and its mechanism using spatial measures and mediating effects. The topic is of high theoretical and practical value, the literature review and theoretical foundation are detailed, and the research methodology is also scientific. Thus, it is recommended to publish after revision. The following suggestions are for the authors to revise and refer to:
(1) The innovation point of the paper is not clearly written out, and it is suggested to add the innovation point in the introduction section.
(2) In the "3.3.2 Explanatory variables" section, the relationship between “First-level indicators” and “Second-level indicators” in "Table 1 Evaluation index system of development level of information infrastructure" is not clearly explained in the text, and it is suggested further clarify it in the text.
(3)In the "5. Conclusions and Discussion" section, the outlook for the future is somewhat weak. "Due to the availability of data, future studies can further increase the time span or obtain more adequate sample data from municipal administrative units to test the reliability of the findings of this study". The paper considers only from the perspective of data acquisition and suggests further strengthening in terms of research perspectives and other aspects.
(4) There are some minor problems in the “Reference” section. For example, the page numbers of reference 1, 9, 10, 11, and 30, etc. are not clearly marked, and further checking is recommended.
Author Response
Reply to Reviewer 1’s comments
Reviewer 1’s comments help us improve our manuscript. We gratefully appreciate and have revised. Following the order of comments, the responses are stated as follows:
Comment 1:
The innovation point of the paper is not clearly written out, and it is suggested to add the innovation point in the introduction section.
Reply 1:
Thanks for this suggestion. According to the suggestions, we noted it in the part of the introduction (see the last paragraph of the introduction).
“The innovation of this paper are as follows: first, the black box of the role of information infrastructure on air quality improvement is unveiled from the national scale by including information infrastructure construction and air quality improvement into a unified research framework; second, the spatial spillover effect of information infrastructure construction on air quality improvement is explored to provide a more precise development direction for the joint prevention and control of regional air pollution from the perspective of information infrastructure construction.”
Comment 2:
In the "3.3.2 Explanatory variables" section, the relationship between “First-level indicators” and “Second-level indicators” in "Table 1 Evaluation index system of development level of information infrastructure" is not clearly explained in the text, and it is suggested further clarify it in the text.
Reply 2:
Thanks for this comment, which makes our expression more detailed and accurate. According to the comment, we did some modifications (shown in “3.3.2 Explanatory variables").
“It includes communication network infrastructure represented by 5G, Internet of Things, Industrial Internet, and satellite Internet; new technology infrastructure represented by artificial intelligence, cloud computing, and blockchain; and computing power infrastructure represented by data centers and intelligent computing centers.”
Comment 3:
In the "5. Conclusions and Discussion" section, the outlook for the future is somewhat weak. "Due to the availability of data, future studies can further increase the time span or obtain more adequate sample data from municipal administrative units to test the reliability of the findings of this study". The paper considers only from the perspective of data acquisition and suggests further strengthening in terms of research perspectives and other aspects.
Reply 3:
Thanks for this suggestion. According to the suggestion, we supplemented it in the part of the conclusion and discussion (see the last paragraph).
“Although this study accomplishes the initial objectives and clarifies the relationship between information infrastructure and air pollution, this paper still has some shortcom-ings. Due to the lack of information infrastructure data, the empirical study only explores the influence mechanism of information infrastructure on air pollution from the provin-cial level, while the influence mechanism is not the same in different spatial scales. It is an innovative idea to consider air pollution from the perspective of information infrastructure. However, air pollution is a complex issue. It may be more necessary to comprehensively analyze air pollution by integrating green technology (e.g. energy-efficient houses, emis-sion control devices etc. [59]), innovative materials (e.g. bamboo as green energetic plant [60], hydrogen [61] etc.), etc. With the improvement of spatial and temporal data, it may become a future research direction to explore the mechanism of information infrastructure influence on spatial pollution based on multiple spatial scales.”
Comment 4:
There are some minor problems in the “Reference” section. For example, the page numbers of reference 1, 9, 10, 11, and 30, etc. are not clearly marked, and further checking is recommended.
Reply 4:
Thanks for this suggestion. According to the suggestion, we have carefully checked the format of the references in this paper and revised some mistakes.
“1. Ministry of Ecology and Environment of the People's Republic of China. China Ecological Environment Status Bulletin 2021. 2021.
- Yang X, Lin S, Li Y, et al. Can high-speed rail reduce environmental pollution? Evidence from China. Journal of Cleaner Production, 2019, 239:118135.
- Park Y, Meng F, Baloch M A. The effect of ICT, financial development, growth, and trade openness on CO2 emissions: an empirical analysis. Environmental Science and Pollution Research, 2018, 25:30708-30719.
- Avom D, Nkengfack H, Fotio H K, et al. ICT and environmental quality in Sub-Saharan Africa: Effects and transmission channels. Technological Forecasting and Social Change, 2020, 155:1-12.
- Wu H, Hao Y, Ren S, et al. Does internet development improve green total factor energy efficiency? Evidence from China. Energy Policy, 2021, 153:1-13.
- Yue H, He C, Huang Q, et al. Stronger policy required to substantially reduce deaths from PM2.5 pollution in China. Nature Communications, 2020, 11(1):1-10.”
Reviewer 2 Report
The paper measures the development level of information infrastructure for 31 provinces in China from 2013 to 2020, using the Entropy‐TOPSIS weight method , examined the impact of information infrastructure on air pollution. The presented results seem to be usefully showing that information infrastructure can effectively improve air quality reducing air pollution and by influencing technological innovation first and then industrial structure upgrading.
Author Response
Reply to Reviewer 2’s comments
Comments to the author
The paper measures the development level of information infrastructure for 31 provinces in China from 2013 to 2020, using the Entropy‐TOPSIS weight method, examined the impact of information infrastructure on air pollution. The presented results seem to be usefully showing that information infrastructure can effectively improve air quality reducing air pollution and by influencing technological innovation first and then industrial structure upgrading.
Reply:
Thanks for your kind comments very much. By exploring the impact of information infrastructure on air pollution and its action path, we expect to provide some scientific reference value for the construction of information infrastructure under the background of the new global technological revolution.
Reviewer 3 Report
Suggessions are marked on pdf file attached and highted in yellow color

Author Response
Reply to Reviewer 3’s comments
Reviewer 3’s comments are constructive in improving our manuscript. We gratefully appreciate and have made a major revision. Following the order of comments, the responses are stated as follows:
Comment 1:
Put full stop here.
Reply 1:
Thanks for this suggestion. According to the suggestion, we have modified this.
“On the other hand, the application of ICT and digitization requires a large amount of energy and materials, as well as the continuous elimination of electronic waste will negatively affect the quality of the environment. Thus, the relationship between information infrastructure and environmental pollution shows an inverted U-shape.”
Comment 2:
Re-write the sentence: “Given this, this paper tries to investigate whether information infrastructure can improve air quality.”
Reply 2:
Thanks for this suggestion. According to the suggestion, we re-write the sentence: “Under this background, this paper attempts to explore whether information infra-structure can improve air quality.”
Comment 3:
Figure 1. Hypothesis 2 is missing here please check. this figure is also not cited in main text. please cite.
Reply 3:
Thanks for this suggestion. We checked. And we added hypothesis 2 to Figure 1 and cited it in the article (see the last paragraph of the “2. Theoretical framework and research hypothesis”).
In summary, this study constructs a theoretical framework for the impact of information infrastructure on air pollution (Figure 1.).
Figure 1. Conceptual framework of the impact of information infrastructure on air pollution
Comment 4:
In section “3.2.1 Entropy-TOPSIS method”, include an example case for this model as an appendix. It will increase the visibility and applicabilities of the current paper.
Reply 4:
Thanks for this suggestion. We added relevant references.
“The Entropy-TOPSIS method is an improvement of the traditional TOPSIS evaluation method [35-37].”
- Vedagiri P, Tekal S, Purna Chandu S. Experimental and numerical studies on biodiesel ternary fuel blends using entropy TOPSIS method. Materials Today: Proceedings, 2022.
- Boafo-Mensah G, Neba F A, Tornyeviadzi H M, et al. Modelling the performance potential of forced and natural-draft biomass cookstoves using a hybrid Entropy-TOPSIS approach. Biomass and Bioenergy, 2021, 150:106106.
- Goswami S S, Jena S, Behera D K. Selecting the best AISI steel grades and their proper heat treatment process by integrated entropy-TOPSIS decision making techniques. Materials Today: Proceedings, 2022, 60:1130-1139.
Comment 5:
Please re-frame the sentence: “high values are adjacent to high values and low values are adjacent to low values.”
Reply 5:
Thanks for this suggestion. We checked the relevant literature and rewrote the sentence.
“I greater than 0 indicates a positive spatial correlation, that is, high-high and low-low clusters.”
“I less than 0 indicates a negative spatial correlation, that is, high-low and low-high outliers.”
Reference:
- Hughey S M, Kaczynski A T, Porter D E, et al. Spatial clustering patterns of child weight status in a southeastern US county. Applied Geography, 2018, 99:12-21.
Comment 6:
Please elaborate “econometric regression method”.
Reply 6:
Thanks for this suggestion, and we are sorry for our inaccurate statement. According to your suggestion, we revised this statement.
“the OLS (ordinary least squares) method”.
Comment 7:
Re-write the sentence “the spatial lags of the explanatory and explanatory variables based on both endogenous and exogenous effects.”
Reply 7:
Thanks for this suggestion, and we are sorry for our inaccurate statement. According to your suggestion, we rewrote this sentence.
“However, compared with the Spatial Lag Model (SLM) and the Spatial Error Model (SEM), the spatial Durbin model (SDM) not only considers the endogenous and exogenous spatial interaction effects but also reflects the spatial hysteresis of explanatory variables and explained variables.”

Reviewer 4 Report
Dear Authors,
Thank you very much for possibility to read your very important paper. After reading your text, I have some comments, advice and suggestions.
Line 37 – please make number for that citation. It should be [1].
Line 39 – what does Internet of everything mean? Maybe Authors wanted to write Internet of Thing (IoT)? The Internet of Things is an element of Industry 4.0, as well as other elements, incl. big data, cloud computing ... I propose to read and support this part of paper with the article: Innovative Processes in Managing an Enterprise from the Energy and Food Sector in the Era of Industry 4.0. Processes. 2021; 9(2):381.
Line 65 and next – In the third way please add some more information about digitization. In energy saving as well as industrial process optimization we use very wide digitization. I suggest to add 2-3 sentences concerning digitization. Please support your paper by some citations e.g: Digitization, Digital Twins, Blockchain, and Industry 4.0 as Elements of Management Process in Enterprises in the Energy Sector. Energies. 2021; 14(7):1885 or/and Energy Is Essential, but Utilities? Digitalization: What Does It Mean for the Energy Sector? In Digital Marketplaces Unleashed; Springer: Berlin/Heidelberg
Line 78 - Given this, this paper – please rewrite that words: this, this
Please try to correct the Figure 1 with Hypotheses H3, H4, H5. E.g. now it can be understand that Information infrastructure has influence on Industrial Structure Upgrading and Industrial Structure Upgrading has influence on Air pollution. Similar comments to other arrow. Where is Hypothesis H2?
The Discussion and Conclusion section is too short when comparing the other parts of the article. I propose to have a separate section “discussion”. The conclusion may be as long as it is now, but please have discussions. In the "discussion" part, please answer the question about what results from your research. Please state whether you have achieved the assumed research goal and verify the initial assumptions. In addition, please link the results of your research with other available results and present the importance of our research for the further development of science.
Please add some papers published in 2021, 2022. In the section where Authors write about reduction of air pollution not only information infrastructure is useful, but also innovative materials used by industry. Authors can find few samples: e.g bamboo as green energetic plant, hydrogen etc.
I hope that my comments are useful for the Authors
Author Response
Reply to Reviewer 4’s comments
Reviewer 4’s comments are constructive in improving our manuscript. We gratefully appreciate and have made a major revision. Following the order of comments, the responses are stated as follows:
Comment 1:
Line 37 – please make number for that citation. It should be [1].
Reply 1:
Thanks for this suggestion. We made number for that citation. “According to the China Ecological Environment Status Bulletin 2021, the average concentration of PM2.5 in 339 cities nationwide was 30µg/m3 in 2021, down about 40% from 2015, but more than 1/3 of cities still failed to meet air quality standards [1], which also means that the situation of air pollution management is still severe in China.”
Comment 2:
Line 39 – what does Internet of everything mean? Maybe Authors wanted to write Internet of Thing (IoT)? The Internet of Things is an element of Industry 4.0, as well as other elements, incl. big data, cloud computing ... I propose to read and support this part of paper with the article: Innovative Processes in Managing an Enterprise from the Energy and Food Sector in the Era of Industry 4.0. Processes. 2021; 9(2):381.
Reply 2:
Thank you very much for this comment. Yes, we wanted to write Internet of Thins (IoT). And we cited the reference “Innovative Processes in Managing an Enterprise from the Energy and Food Sector in the Era of Industry 4.0”, and replaced “Internet of everything” with “Internet of Thing (IoT)”.
“However, in today's era of the Industry 4.0, with the development of machine learning, Internet of Thing (IoT), Big Data, or augmented reality [2], information infrastructure is more capable of breaking the bottleneck of air pollution control and achieving air quality improvement than transportation infrastructure that achieves air pollution reduction under virtue of spatial and temporal compression effect [3,4].”
- Borowski P. Innovative Processes in Managing an Enterprise from the Energy and Food Sector in the Era of Industry 4.0. Processes, 2021, 9:381.
- Yang X, Lin S, Li Y, et al. Can high-speed rail reduce environmental pollution? Evidence from China. Journal of Cleaner Production, 2019, 239:118135.
- Dalkic G, Balaban O, Tuydes-Yaman H, et al. An assessment of the CO2 emissions reduction in high speed rail lines: Two case studies from Turkey. Journal of Cleaner Production, 2017, 165:746-761.
Comment 3:
Line 65 and next – In the third way please add some more information about digitization. In energy saving as well as industrial process optimization we use very wide digitization. I suggest to add 2-3 sentences concerning digitization. Please support your paper by some citations e.g: Digitization, Digital Twins, Blockchain, and Industry 4.0 as Elements of Management Process in Enterprises in the Energy Sector. Energies. 2021; 14(7):1885 or/and Energy Is Essential, but Utilities? Digitalization: What Does It Mean for the Energy Sector? In Digital Marketplaces Unleashed; Springer: Berlin/Heidelberg
Reply 3:
Thank you very much for this comment, which makes our writing clearer. We added 3 sentences concerning digitization and cited 5 references.
“Third, there is a non-linear relationship between information infrastructure construction and air pollution. On the one hand, information infrastructure promotes the development of information technology and digitalization [16]. The information and communications technology (ICT) can achieve energy saving and emission reduction through the development of smart cities, transportation systems, and industrial process optimization [17]. Digitization gives the opportunity to increase energy efficiency of energy companies [18,19]and plays a central role in the transition of the energy sector [20], so it improves green total factor energy efficiency [21]. On the other hand, the application of ICT and digitization requires a large amount of energy and materials, as well as the continuous elimination of electronic waste will negatively affect the quality of the environment. Thus, the relationship between information infrastructure and environmental pollution shows an inverted U-shape.”
- Zhou Y, Xiao F, Deng W. Is smart city a slogan? Evidence from China. Asian Geographer, 2022:1-18.
- Añón Higón D, Gholami R, Shirazi F. ICT and environmental sustainability: A global perspective. Telematics and In-formatics, 2017, 34(4):85-95.
- Borowski P F. Digitization, Digital Twins, Blockchain, and Industry 4.0 as Elements of Management Process in Enterprises in the Energy Sector. Energies. 2021.
- Trahan R T, Hess D J. Who controls electricity transitions? Digitization, decarbonization, and local power organizations. Energy Research & Social Science, 2021, 80:102219.
- Varela I. Energy Is Essential, but Utilities? Digitalization: What Does It Mean for the Energy Sector? 2018: 829-838.
- Gao D, Li G, Yu J. Does digitization improve green total factor energy efficiency? Evidence from Chinese 213 cities. Energy, 2022, 247:123395.
Comment 4:
Line 78 - Given this, this paper – please rewrite that words: this, this
Reply 4:
Thanks for this suggestion. According to the suggestion, we re-write the sentence: “Under this background, this paper attempts to explore whether information infra-structure can improve air quality.”
Comment 5:
Please try to correct the Figure 1 with Hypotheses H3, H4, H5. E.g. now it can be understand that Information infrastructure has influence on Industrial Structure Upgrading and Industrial Structure Upgrading has influence on Air pollution. Similar comments to other arrow. Where is Hypothesis H2?
Reply 5:
Thanks for this suggestion. We checked the Figure 1. And we added hypothesis 2 to Figure 1 and cited it in the article (see the last paragraph of the “2. Theoretical framework and research hypothesis”).
In summary, this study constructs a theoretical framework for the impact of information infrastructure on air pollution (Figure 1.).
Figure 1. Conceptual framework of the impact of information infrastructure on air pollution
Comment 6:
The Discussion and Conclusion section is too short when comparing the other parts of the article. I propose to have a separate section “discussion”. The conclusion may be as long as it is now, but please have discussions. In the "discussion" part, please answer the question about what results from your research. Please state whether you have achieved the assumed research goal and verify the initial assumptions. In addition, please link the results of your research with other available results and present the importance of our research for the further development of science.
Reply 6:
Thank you very much for this comment, which makes our writing more reasonable. We added a separate section “discussion”. According to the comment, we answered the question about what results from this study and tried to link the results of this paper with other available results and present the importance of our research for the further development of science (see in the "discussion" part).
“5.2 Discussion
The findings of this paper further confirm that the construction of information infrastructure can improve air quality and reveal its intrinsic mechanisms. Also, this paper tests that factors such as economic growth [48], population density [50], green area [23], and urbanization [57] can also affect air pollution. Combining the above findings, this paper draws the following policy insights: at the macro level, the national government can introduce relevant policies to encourage the construction of information infrastructure, such as financial subsidies, etc. At the micro level, each region should take into account its comparative advantages and promptly learn from the beneficial experiences of other provinces across the country, such as how information infrastructure can promote industrial structure upgrading, to actively improve air pollution levels.
Although this study accomplishes the initial objectives and clarifies the relationship between information infrastructure and air pollution, this paper still has some shortcomings. Due to the lack of information infrastructure data, the empirical study only explores the influence mechanism of information infrastructure on air pollution from the provincial level, while the influence mechanism is not the same in different spatial scales. It is an innovative idea to consider air pollution from the perspective of information infrastructure. However, air pollution is a complex issue. It may be more necessary to comprehensively analyze air pollution by integrating green technology (e.g energy-efficient houses, emission control devices etc. [58]), innovative materials (e.g bamboo as green energetic plant [59], hydrogen [60] etc.), etc. With the improvement of spatial and temporal data, it may become a future research direction to explore the mechanism of information infrastructure influence on spatial pollution based on multiple spatial scales.”
Comment 7:
Please add some papers published in 2021, 2022. In the section where Authors write about reduction of air pollution not only information infrastructure is useful, but also innovative materials used by industry. Authors can find few samples: e.g bamboo as green energetic plant, hydrogen etc.
Reply 7:
Thank you very much for this comment. According to the comment, we added some papers published in 2021, 2022 (see in the references). At the same time, we also agreed that reduction of air pollution not only information infrastructure is useful, but also innovative materials used by industry. So we believed that future researches may be more necessary to comprehensively analyze air pollution by integrating green technology, innovative materials, etc.
“1. Ministry of Ecology and Environment of the People's Republic of China. China Ecological Environment Status Bulletin 2021. 2021.
- Borowski P. Innovative Processes in Managing an Enterprise from the Energy and Food Sector in the Era of Industry 4.0. Processes, 2021, 9:381.
- Zhou Y, Xiao F, Deng W. Is smart city a slogan? Evidence from China. Asian Geographer, 2022:1-18.
- Borowski P F. Digitization, Digital Twins, Blockchain, and Industry 4.0 as Elements of Management Process in Enterprises in the Energy Sector. Energies. 2021.
- Trahan R T, Hess D J. Who controls electricity transitions? Digitization, decarbonization, and local power organizations. Energy Research & Social Science, 2021, 80:102219.
- Gao D, Li G, Yu J. Does digitization improve green total factor energy efficiency? Evidence from Chinese 213 cities. Energy, 2022, 247:123395.
- Vedagiri P, Tekal S, Purna Chandu S. Experimental and numerical studies on biodiesel ternary fuel blends using entropy TOPSIS method. Materials Today: Proceedings, 2022.
- Boafo-Mensah G, Neba F A, Tornyeviadzi H M, et al. Modelling the performance potential of forced and natural-draft biomass cookstoves using a hybrid Entropy-TOPSIS approach. Biomass and Bioenergy, 2021, 150:106106.
- Goswami S S, Jena S, Behera D K. Selecting the best AISI steel grades and their proper heat treatment process by integrated entropy-TOPSIS decision making techniques. Materials Today: Proceedings, 2022, 60:1130-1139.
- Song Y, Zhang X, Zhang M. The influence of environmental regulation on industrial structure upgrading: Based on the strategic interaction behavior of environmental regulation among local governments. Technological Forecasting and Social Change, 2021, 170:1-12.
- Sarfraz M. Green Technologies to Combat Air Pollution //Tiwari S, Saxena P. Air Pollution and Its Complications: From the Regional to the Global Scale. Cham; Springer International Publishing. 2021: 143-161.
- Zhao K, Ren C, Lu Y, et al. Cellulose nanofibril/PVA/bamboo activated charcoal aerogel sheet with excellent capture for PM2.5 and thermal stability. CARBOHYDRATE POLYMERS, 2022, 291.
- Lewis A C. Optimising air quality co-benefits in a hydrogen economy: a case for hydrogen-specific standards for NOx emissions. Environmental Science: Atmospheres, 2021, 1(5):201-207.”

Round 2
Reviewer 4 Report
The authors took all my comments and suggestions into account, so the paper is now ready for publication. The research hypotheses in the figure are presented in a logical way. The discussion is quite short, but it covers all the relevant issues. I would suggest that the Authors read articles on the development of bamboo as a construction or energy material. Eg Innovative Industrial Use of Bamboo as Key “Green” Material. Sustainability, (2022), 14 (4), 1955. Bamboo is an interesting, innovative and perspective material, and the authors will look at bamboo in a broader way.
In the conclusion section please write 1-2 sentences what is the contribution (addded value) for the readers.
Author Response
We are very grateful for the time and efforts of the Editorial Board and the Reviewers in reviewing our manuscript. We have revised the manuscript to improve its quality and address the specific suggestions and constructive comments of the reviewer.
Comments:
The authors took all my comments and suggestions into account, so the paper is now ready for publication. The research hypotheses in the figure are presented in a logical way. The discussion is quite short, but it covers all the relevant issues. I would suggest that the Authors read articles on the development of bamboo as a construction or energy material. Eg Innovative Industrial Use of Bamboo as Key “Green” Material. Sustainability, (2022), 14 (4), 1955. Bamboo is an interesting, innovative and perspective material, and the authors will look at bamboo in a broader way.
In the conclusion section please write 1-2 sentences what is the contribution (added value) for the readers.
Reply:
Thank you for your constructive and specific comments. Your valuable suggestion relates to the novelty of our research which is critical important. We read the reference you recommended and added it in the references section.
- Innovative Industrial Use of Bamboo as Key “Green” Material. Sustainability, 2022, 14 (4), 1955.
And we wrote 2 sentences what is the contribution for the readers in the conclusion section.
“This paper explores the impact of information infrastructure on air pollution and its action path, which has a very important theoretical and practical significance. It not only could provide theoretical support for the information infrastructure to improve air quality, but also might offer scientific reference for the construction and layout of information infrastructure in the context of the new round of global technological revolution.”
We hope the above responses are adequate and the revisions are acceptable. Thanks a lot for the reviewer’s valuable comments and kind efforts.
